# Network Modeling of Murine Lymphatic System

**Dmitry Grebennikov** [1,2,3,*], **Rostislav Savinkov** [1,2,3], **Ekaterina Zelenova** [4], **Gennady Lobov** [5] **and Gennady Bocharov** [1,2,3,*]

1 Marchuk Institute of Numerical Mathematics, Russian Academy of Sciences, 119333 Moscow, Russia; dr.savinkov@gmail.com
2 Moscow Center for Fundamental and Applied Mathematics at INM RAS, 119333 Moscow, Russia
3 Institute of Computer Science and Mathematical Modelling, Sechenov First Moscow State Medical University, 119991 Moscow, Russia
4 Moscow Institute of Physics and Technology, National Research University, 141701 Dolgoprudny, Russia; kdzelenova@gmail.com
5 Pavlov Institute of Physiology, Russian Academy of Sciences, 119034 Saint-Petersburg, Russia; gilobov@yandex.ru
* Correspondence: grebennikov_d_s@staff.sechenov.ru (D.G.); g.bocharov@inm.ras.ru (G.B.)

**Abstract:** Animal models of diseases, particularly mice, are considered to be the cornerstone for translational research in immunology. The aim of the present study is to model the geometry and analyze the network structure of the murine lymphatic system (LS). The algorithm for building the graph model of the LS makes use of anatomical data. To identify the edge directions of the graph model, a mass balance approach to lymph dynamics based on the Hagen–Poiseuille equation is applied. It is the first study in which a geometric model of the murine LS has been developed and characterized in terms of its structural organization and the lymph transfer function. Our study meets the demand for quantitative mechanistic approaches in the growing field of immunoengineering to utilize or exploit the lymphatic system for immunotherapy.

**Keywords:** lymphatic system; experimental mice; network; graph model; topology; computation; lymph flow





## 1. Introduction

Networks of various natures, e.g., structural, functional, spatial, underlie the dynamics and mechanisms of regulation in live systems ranging from cells to physiological organs and to whole organisms [1]. Network concepts are increasingly used to describe the structure and function of the immune system [2]. The immune system represents an example of a highly complex network of interacting and migration cell populations embedded into the spatially distributed components of the lymphatic system (LS) [3].

The LS is a body-wide network of lymphatic vessels and lymphoid organs with two major functions: (1) fluid transport from tissues to the blood system to maintain fluid homeostasis and (2) trafficking of antigens and immune cells to lymph nodes where the immune responses take place [4,5]. Lymphoid organs include a large number of lymph nodes as well as the spleen, thymus, tonsils, and bone marrow [5]. The lymphatic vessels are the conduits that facilitate the directional lymph transport from peripheral tissues to secondary lymph nodes [4]. Understanding of the lymphatic structural and functional organization is essential to discern how the LS interacts with different tissues and organs within the body [6].

Animal models of diseases, particularly mice, are considered to be the cornerstone for translational research in immunology [7,8]. Research with laboratory mice enabled invaluable insight into mammalian immune systems [9]. Despite numerous advances in understanding the immune system from mouse studies, there exist fundamental differences between mouse and human immune systems [8]. The structural organization of

the lymphatic system represents a straightforward example. However, a comprehensive mathematical network-based characterization of the LS in mice is still unavailable.

The primary function of the lymphatic system is the maintenance of the interstitial fluid homeostasis [10]. The structure and topology of the LS network is heterogeneous and remains to be systematically explored [5]. The existing mathematical models of the lymphatic system refer to specific parts of it, such as the lymphatic capillary network [11], collecting lymphatics [12], lymphangions [13], or branching networks of lymphatic vessels [14]. Computational models of the whole lymphatic system network are still rare [15–18]. Comprehensive reviews of existing approaches to modeling the lymphatic system structure and function can be found in [10,19]. One of the major bottlenecks in developing the computational models of the lymphatic system is due to the lack of comprehensive anatomical and physiological data [10]. The latest research activity has clearly stated, "Thus, gross lymphatic anatomy has not been updated for more than a century. ... our knowledge of macro-lymphatic anatomy remains rudimentary" [6]. The existing gaps [20], require further systematic research [21] including mathematical modeling [22], which serves to integrate available knowledge and to identify critical issues amenable to further biological testing.

The application of graph theory methods to describe the spatial organization of the human LS has been addressed in a number of recent studies (see for a review [18]). In [18], we developed a computational algorithm for representing the anatomy-based and rule-based graphs of the LS in humans. The graph models enabled the analysis of different metrics of complexity of the human LS such as spectral radius, clustering coefficient, average path length, and number of separators. A similar analysis for the mouse LS remains to be performed.

The aim of the present study is to model and analyze the network structure of the murine lymphatic system. The algorithm for building the graph model makes use of anatomical data. To identify the edge directions of the graph model, a mass balance approach to lymph dynamics based on the Hagen–Poiseuille equation is applied. Various matrix forms for graph representation are specified. The lymph transfer times between various nodes are estimated. We summarize the properties of the graph model of the murine LS using metrics similar to the human LS graph thus providing a quantitative basis for understanding essential structural differences of the LS between the mice and humans.

## 2. Anatomy and Physiology of Murine Lymphatic System

Available anatomical and physiological data provide the empirical basis for specifying the network structure of the lymphatic system in mice [23,24] using the notion of a simple graph. A simple graph $G = (V, E)$ is a pair of sets $V$ and $E$, with elements of $V$ being vertices or nodes and $E$ being edges [25]. The simple graph with edges oriented in only one direction is called an oriented graph. There are some variations in the number of lymph nodes, i.e., ranging from 22 to 36 as indicated in Table 1. A generalized graph of the murine LS, consisting of 88 nodes and 87 edges is shown in Figure 1. It was developed using anatomical descriptions from [23,24]. The vertices of the graph refer to either the lymph nodes, outlet vertices with out-degree $deg^+ = 0$ corresponding to the sink into jugular veins, the confluences of lymphatic vessels, or inlet vertices with in-degree $deg^- = 0$ corresponding to the collecting lymphatics of various body tissues.

The geometric characteristics of the lymphatic vessels and the baseline parameters of the lymph flow through various parts of the LS network are detailed in Table 1. To set the pressure at the sink nodes $p_{out}$, we used the estimate of the murine central venous pressure: 7.4 (5.9–8.9) cm $H_2O$ [26,27]. Lymph viscosity is taken to be equal to 1.81 mPa·s.

The anatomy data enable specifying a simple graph of the murine LS. The adjacency matrix $A$ of the LS graph is shown in Figure 2B.

As the LS functions to transport the lymph from the drained tissues to the venous part of cardiovascular system, additional analysis is required to transfer the simple graph representation to a physiologically meaningful oriented graph of the LS. To generate an oriented graph mode of the LS, we used a combination of experimental stud-

ies on fluid dynamics in various parts of the LS in mice [28–31] and computation of the lymph flow through the system in accordance to an overall mass balance using the Hagen–Poiseuille equations.

**Table 1.** Physiological and anatomical properties of the murine lymphatic system.

| Property | Characteristic Value/Range | Commentaty/Source |
|---|---|---|
| Lymph nodes: | | |
| Number | 22 | (BALB/cAnNCr) [24] |
| | 28–36 | (DD/NIH) [23] |
| Diameter | 1–2.3 mm | (C57Bl/6J, Nude, CB-17 SCID) [31,32] |
| | 1–17.3 mm | (DD/NIH) [23] |
| Thoracic duct: | | |
| Radius | 300 μm | [28] |
| Flow | 417–1250 μL/h | (10 mL/day for immobilized mice, 30 mL/day after movements) [28] |
| Velocity | 410–1228 μm/s | [28] |
| Vessels afferent to popliteal nodes: | | |
| Radius | 20–40 μm | [28,30] |
| Flow | 0.3–3.4 μL/h | (mean flow = 0.3 μL/h [30]) [28–30] |
| Velocity | 37–186 μm/s | (mean velocity = 53 ±16 μm/s [30]) [28–30] |
| Collecting lymphatics in hind limbs: | | |
| Velocity | 50–100 μm/s | [31] |
| Collecting lymphatics in ears: | | |
| Velocity | 0–400 μm/s | [30] |
| Collecting lymphatics in the tail: | | |
| Velocity | 4.2 μm/s | [29] |

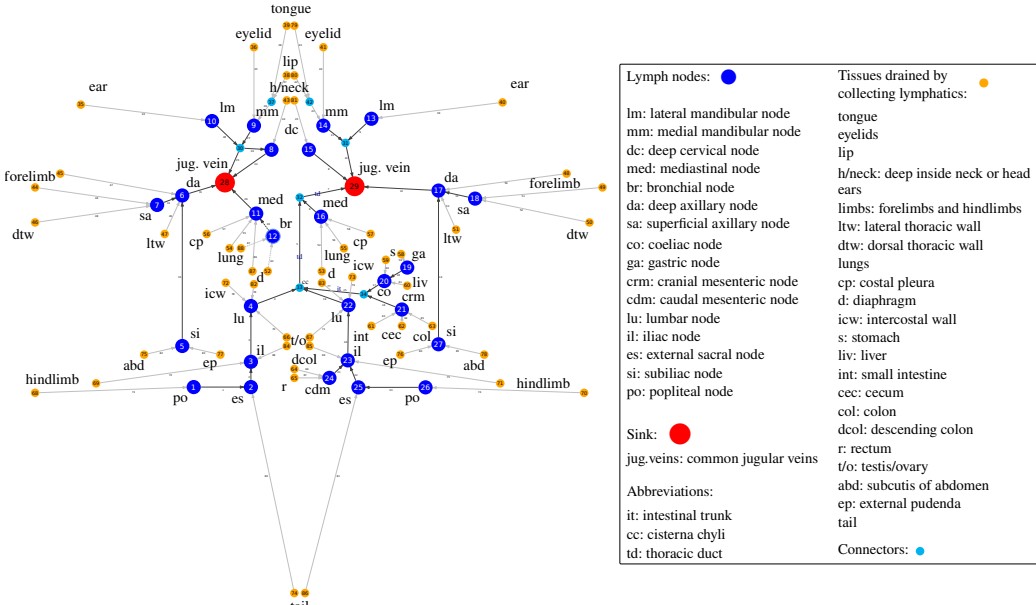

**Figure 1.** Oriented graph of the murine lymphatic system based on the anatomic data with 88 vertices and 87 edges. The vertices of the graph belong to four groups as detailed in the legend box: (1) lymph nodes (large blue), (2) outlet vertices with out-degree $deg^+ = 0$ corresponding to the sink into jugular veins (red), (3) connectors, i.e., the confluences of lymphatic vessels (light blue), (4) inlet vertices with in-degree $deg^- = 0$ corresponding to the collecting lymphatics of various body tissues (orange). The vertex IDs and edge IDs are enumerated arbitrarily for correspondence with the matrix representation of the graph in Figure 2. The presented graph is specified in the CSV files containing the vertex and edge lists in Supplementary Materials.

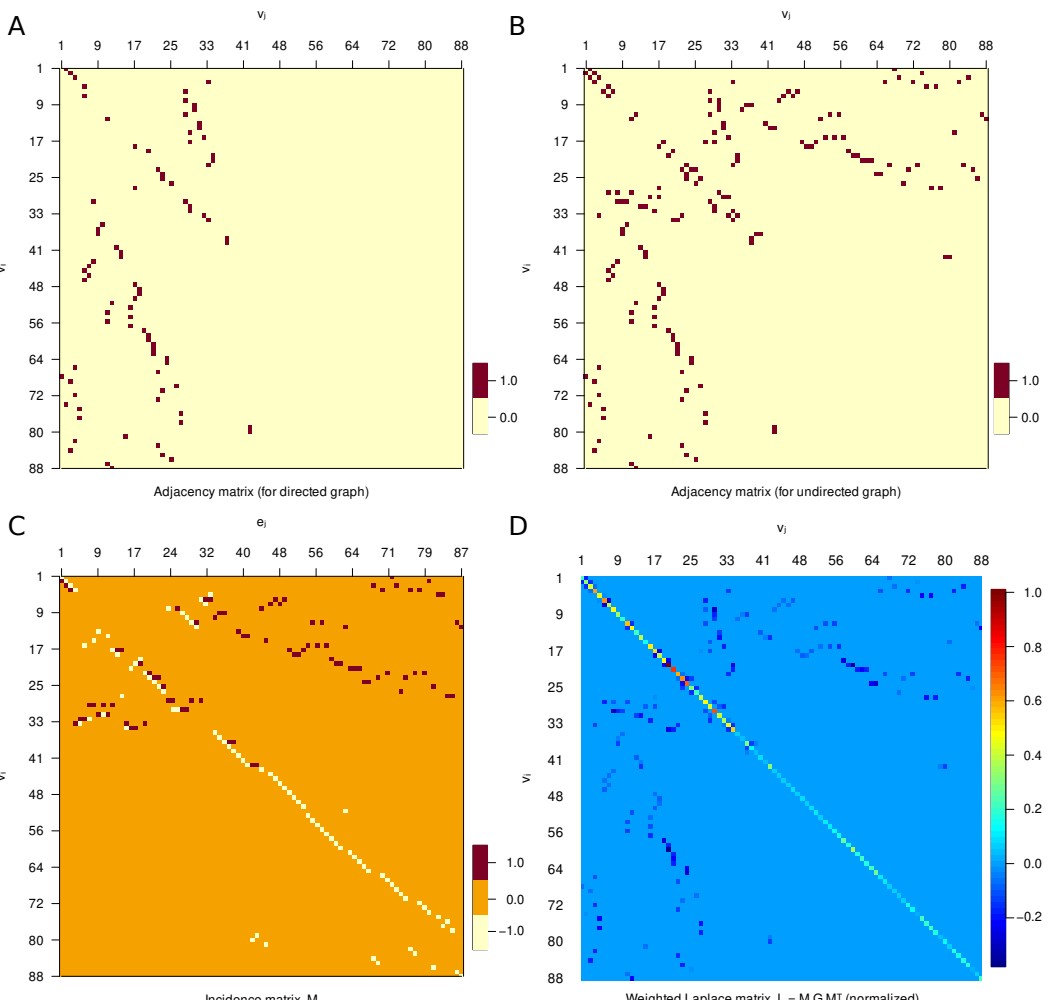

**Figure 2.** Matrix representation of the anatomy-based graph of murine lymphatic system presented in Figure 1. The *i*-th and *j*-th vertices and edges are denoted as $v_i$, $v_j$ and $e_i$, $e_j$, respectively. (**A,B**) Adjacency matrix $A$ (for oriented and simple graph). (**C**) Incidence matrix, $M$. (**D**) Weighted Laplacian matrix, $L = MGM^T$, normalized by the maximum matrix element. The conductance matrix $G$ for constant vessel diameters is used for the illustration.

The initial collecting lymphatics are likely to differ in the inflows as they absorb lymph from interstitial space characterized by volume and pressure varying across the body. However, the respective anatomical and physiological data to quantify the local impedance of the related edges are largely missing. Hence, we used a simplifying assumption that the flow velocities at all inlet vertices are the same and the pressure at the two sink nodes are equal. The following values of model parameters were used in computations of lymph flows:

- number of inlet vertices $n_{in} = 52$;
- vessel radii range $r_{ij} = 40$–$300$ μm;
- vessel length $l_{ij} = 7$–$60$ mm;
- pressure at the sink nodes $p_{out} = 725$ Pa;
- lymph viscosity $\mu = 1.81$ mPa·s;
- lymph inflow $q_{in} = 0.4$ mL/day.

## 3. Oriented Graph Model of Murine LS

The graph of the lymphatic system $G$ can be divided in two disconnected subgraphs: collecting the lymph to the left common jugular vein ($g_l$) and to the right common jugular vein ($g_r$) (Figures 1 and 3). The thoracic duct belongs to the subgraph $g_l$.

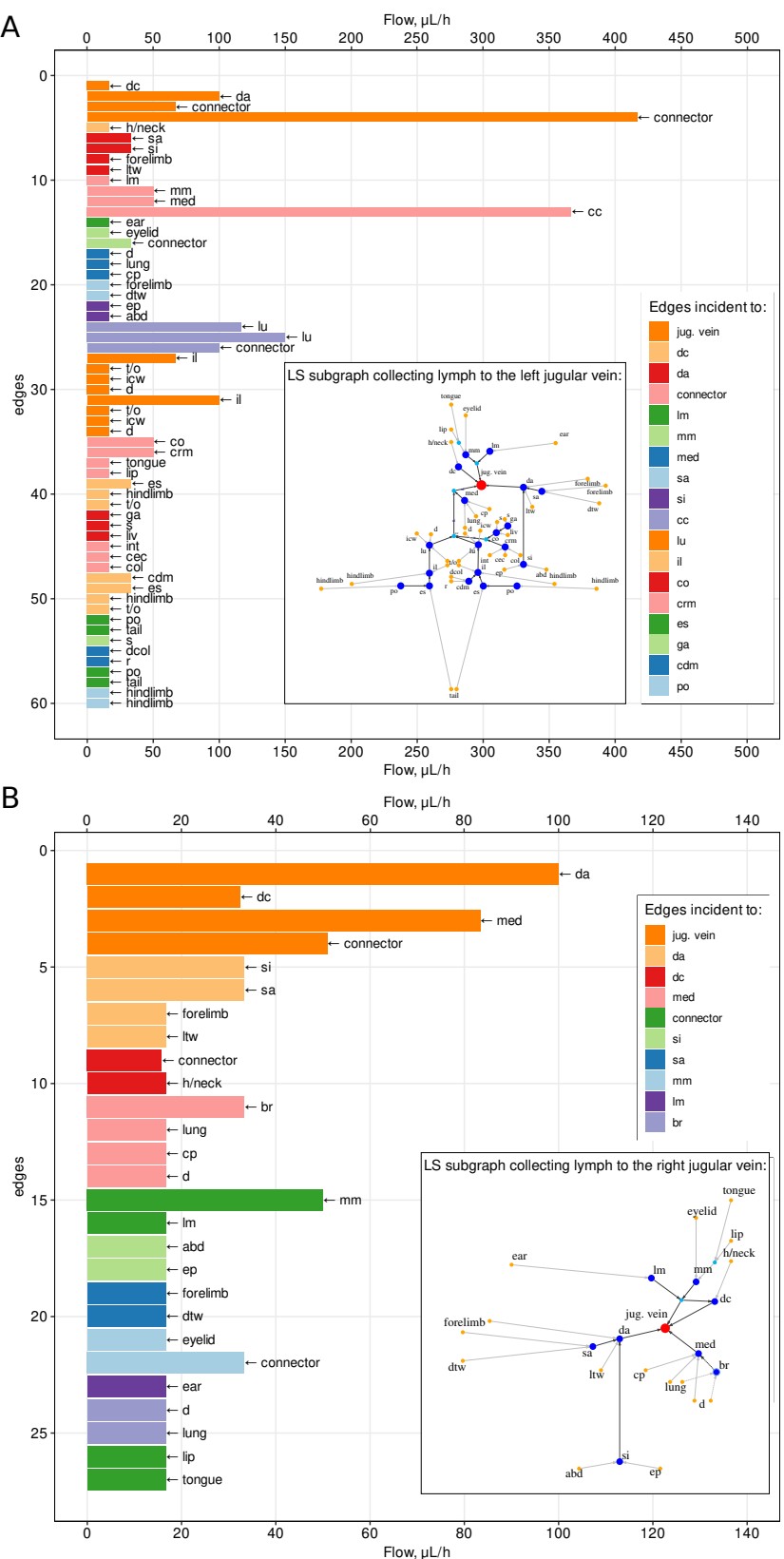

**Figure 3.** The distribution of the flows in the lymphatic vessels for Scenario 2, in which vessels radii decrease linearly with distance from the outlet vertices. Flows are shown in the subgraphs of the lymphatics collecting lymph into the left (**A**) and into the right (**B**) jugular vein. Colors and the arrows indicate the heads and the tails of the edges correspondingly. Edges are sorted by their distance from the sink.

### 3.1. Computing the Direction of Lymph Flows

As we aim to reproduce the target flow through thoracic duct $q_{td} = 10$ mL/day (Table 1), we compute the flows in the left subgraph first. The pressure at the sink vertex is set to $p_{out} = 7.4$ cm $H_2O$, and the outflow from the left LS subgraph to the left jugular vein $q_{out}^{(l)}$ is calibrated so that the computed flow through thoracic duct (the edge incident to jugular vein) is equal to $q_{td}$. The inflows in the collecting lymphatics (inlet vertices with zero in-degree) are given to be the same and equal to $q_{in}^{(l)} = q_{out}^{(l)}/n_{in}^{(l)}$, where $n_{in}^{(l)}$ is the number of inlet vertices. After obtaining the flows in the left subgraph $g_l$, we compute the flows in the right subgraph $g_r$ by setting the same inflows $q_{in}^{(r)} = q_{in}^{(l)}$ as in the left one, and the same output pressure $p_{out}$. The outlet outflow is given by $q_{out}^{(r)} = q_{in}^{(r)} n_{in}^{(r)}$.

The distribution of the steady flows in the graph $g(n, m)$ with $n$ vertices and $m$ edges is considered to be governed by the following:

- The Hagen–Poiseuille equation

$$q_{ij} = g_{ij}(p_i - p_j) = \frac{\pi r_{ij}^4}{8\mu l_{ij}}(p_i - p_j), \tag{1}$$

  links the flow $q_{ij}$ through the edge $e_{ij}$ with the drop of pressure from the tail $i$ to the head $j$ vertices ($p_i - p_j$) by the conductances $g_{ij}$ that depend on lymph viscosity $\mu$ and the radii and the lengths of the edges;

- The balance of flow through the vertices due to mass conservation:

$$\sum_{j \in \mathcal{A}(i)} q_{ij} = \begin{cases} q_{in}, & \text{if } i \text{ is inlet vertex} \\ -q_{out}, & \text{if } i \text{ is outlet vertex} \\ 0, & \text{otherwise} \end{cases} \tag{2}$$

  where $\mathcal{A}(i)$ is a set of vertices adjacent to $i$.

Using the oriented incidence matrix $M \in \mathbb{R}^{n \times m}$ and the diagonal conductance matrix $G \in \mathbb{R}^{m \times m}$ (with elements indexed by edges rather than nodes), one can rewrite Equations (1) and (2) as

$$q = -GM^T p, \quad Mq = -\hat{q}, \tag{3}$$

where $p \in \mathbb{R}^n$ are the nodal pressures, $\hat{q} \in \mathbb{R}^n$ are the net flows through the vertices, and $q \in \mathbb{R}^m$ are the flows through the edges. Hence, we get the linear system to solve for the nodal pressures:

$$MGM^T p = Lp = \hat{q}, \tag{4}$$

where $L = MGM^T$ is a symmetric weighted Laplacian matrix.

As at the outlet vertex the pressure is known ($p_{out}$), we substitute the vector $\hat{p} = [0, \dots, p_{out}, \dots, 0]^T$ of zeros with the known pressure at the corresponding index in (4). By shifting $L\hat{p}$ to the right-hand side, we obtain the system

$$L_{rect} p_{unknown} = \hat{q} - L\hat{p}, \tag{5}$$

where $L_{rect}$ is the matrix L without the column corresponding to the index of the outlet vertex with known pressure. The pseudo-inverse of $L_{rect}$ provides the vector of unknown pressures: $p_{unknown} = L_{rect}^+(\hat{q} - L\hat{p})$.

The graph was constructed and visualized using the R package *igraph*. The algorithm for computing the flows described in Section 3.1 was implemented in R using the *ginv()* function for pseudo-inverse calculation from the MASS package. In addition, we have verified the computation of the flows in Julia language using the singular value decomposition to obtain the pseudo-inverse.

An oriented graph of the murine LS resulting from the analysis of the global lymphatic flow balance is shown in Figure 1. It is derived assuming constant diameter of vessels in the LS.

### 3.2. Matrix Representation

To visualize the graph structure of the murine LS, we use the adjacency matrix, which indicates whether a pair of nodes are adjacent or not. For the oriented graph, the adjacency matrix is shown in Figure 2A. The adjacency matrix of the simple graph presented in Figure 2B is symmetric. A complementary representation of the graph is provided by the incidence matrix (see Figure 2C). The incidence matrix is different from an adjacency matrix, and it encodes the relation of node–vertex pairs. Finally, the graph Laplacian matrix is displayed in Figure 2D. It is related to the degree matrix $D$ and the adjacency matrix $A$ of the graph $L = D - A$, representing an edge-weighted graph.

## 4. Quantitative Characterization of Lymph Flows through the LS

The estimated values of the lymph flow through various vessels of the murine LS are specified in Figure 3. The upper panel shows the flow intensity in the major section of the LS, draining the left and lower parts of the body. The baseline values vary from about 20 to 420 μL/h. The lower panel characterizes the flow intensity in the minor section of the LS, draining the upper right part of the body. The baseline values vary from about 20 to only 100 μL/h.

The radius of the thoracic duct is known to be around 300 μm, while the radius of the lymphatic vessels afferent to popliteal nodes is around 20–40 μm (Table 1). For the human LS it is known that the largest lymphatic vessels have a diameter of about 2 mm and the diameter reduces to approximately 10–60 μm for initial lymphatic capillaries [10]. Due to the lack of detailed anatomical and physiological data in mice on the diameters of all lymphatic vessels, we explored three complementary assumptions on the distribution of the radii of the edges of the lymphatic graph, as specified in the following three scenarios:

- Scenario 1. All radii in the graph are assumed to be the same, equal to 150 μm (half of the radius on the thoracic duct).
- Scenario 2. Edge radii decrease linearly with distance from the outlet vertices (jugular veins) to the inlet vertices. On the thoracic duct, the radius is assumed to be 300 μm, on the most distant edges (from the hindlimbs) it is assumed to be 41 μm. Therefore, on other edges from the inlet vertices, the value of the radius is equal to 41 μm and increases linearly when approaching the vein. On the subgraph collecting lymph into the right jugular vein, the radii are set symmetrically, equal to the radii in the left subgraph.
- Scenario 3. Edge radii are distributed so that the cross-sectional area of incoming and outgoing vessels for each vertex of the graph is preserved. On all inlet edges, the radii are assumed to be the same and are estimated so that the radius on the thoracic duct would be equal to 300 μm.

The histograms of the vessel radii distribution for the above scenarios are presented in the left column of Figure 4. They clearly indicate that the median value of the vessel lumen decreases as we move from the first to the third scenario (from 150 μm (Scenario 1), to 115 μm (Scenario 2), to 60 μm (Scenario 3)). Note, that the estimated lymph flows shown in Figure 3 refer to Scenario 2.

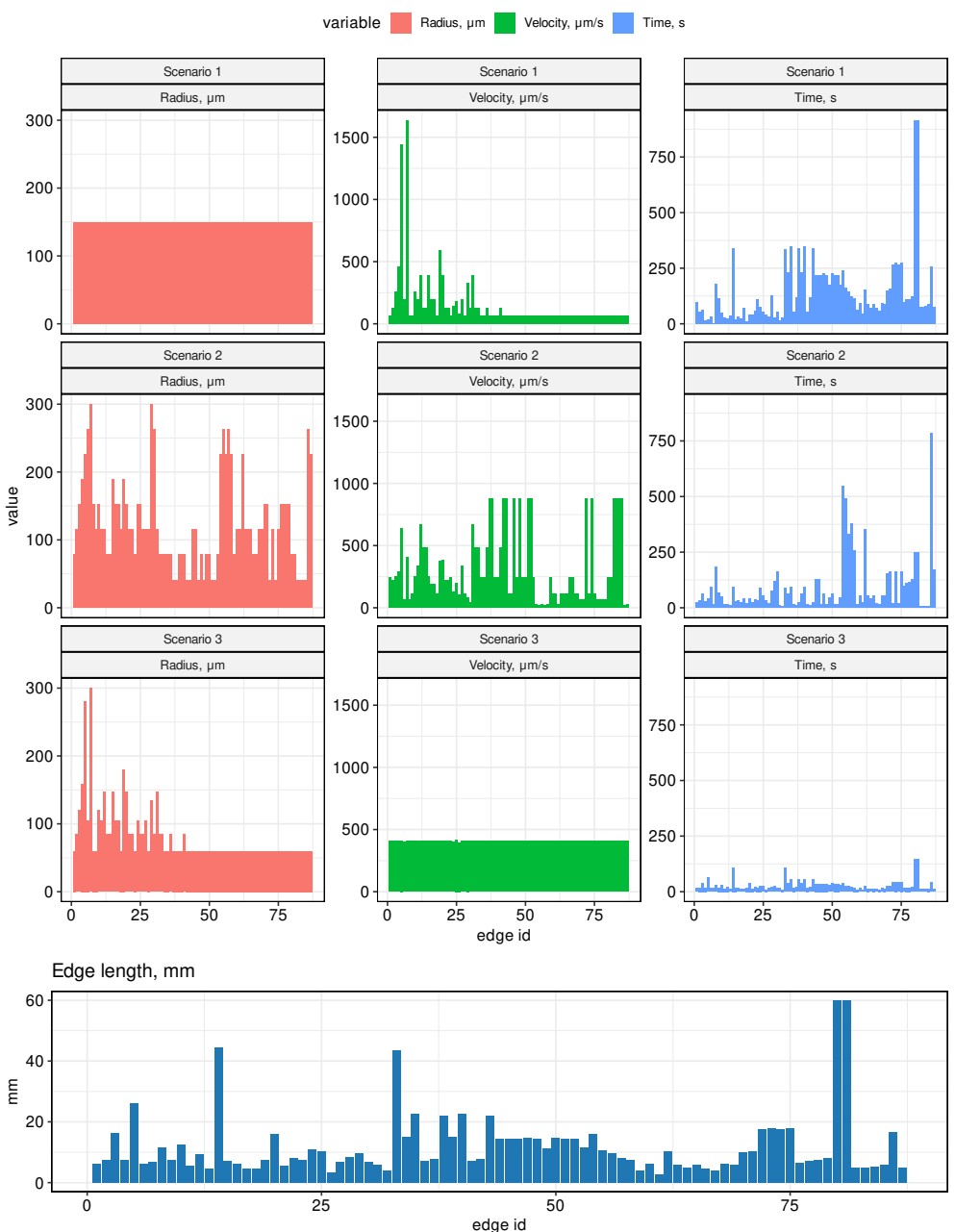

**Figure 4.** The distribution of the velocities and transient transfer times of lymph flow in the vessels for the three scenarios of vessel radii distribution. At the bottom, the distribution of the edge lengths is shown, which is the same for all scenarios.

### 4.1. Lymph Transfer Rates between Lymph Nodes

The key characteristic of the LS function is the rate of transfer of fluid through the system. Using the developed graph model, we estimated the lymph flow rates and the transition times between the lymph nodes. The computational results for three scenarios are summarized in Figure 4. The central column provides the estimates of the flow velocity. The median values turn out to be the smallest for the scenario of uniform vessel diameters and the largest under the assumption of conservation of the cross-sectional area at the vessel junctions. In particular, it increases from about 66 µm/s (Scenario 1), to 242 µm/s (Scenario 2), to 409 µm/s (Scenario 3). In addition, it is predicted to be practically homogeneous across the LS in the third case. The respective median transfer times between neighboring

edges increase from about 19 s to 95 s with the longest transfer times from 2 min 25 s to 15 min 15 s when we compare the third and the first scenarios of the vessel geometry.

### 4.2. Sensitivity to Variations in Vessels Diameter

To analyze the sensitivity of the system to the diameter of the vessels, for all three scenarios, situations were simulated when the diameter of the vessel represented by the corresponding branch of the graph was varied by +10% and −10% from the initial one. The effect of change of the vessels' diameters was expressed in terms of the histograms of the relative change of pressure in the vessels of the LS as shown in Figure 5. In all three scenarios, the reduction of the radii led expectedly to a pressure increase, while the increase of the vessel radii had an opposite effect. The degree of variation was smallest for Scenario 1 and was largest for Scenario 2. The 10% diameter variation led to less than 1% change in the pressure for most of the vessels.

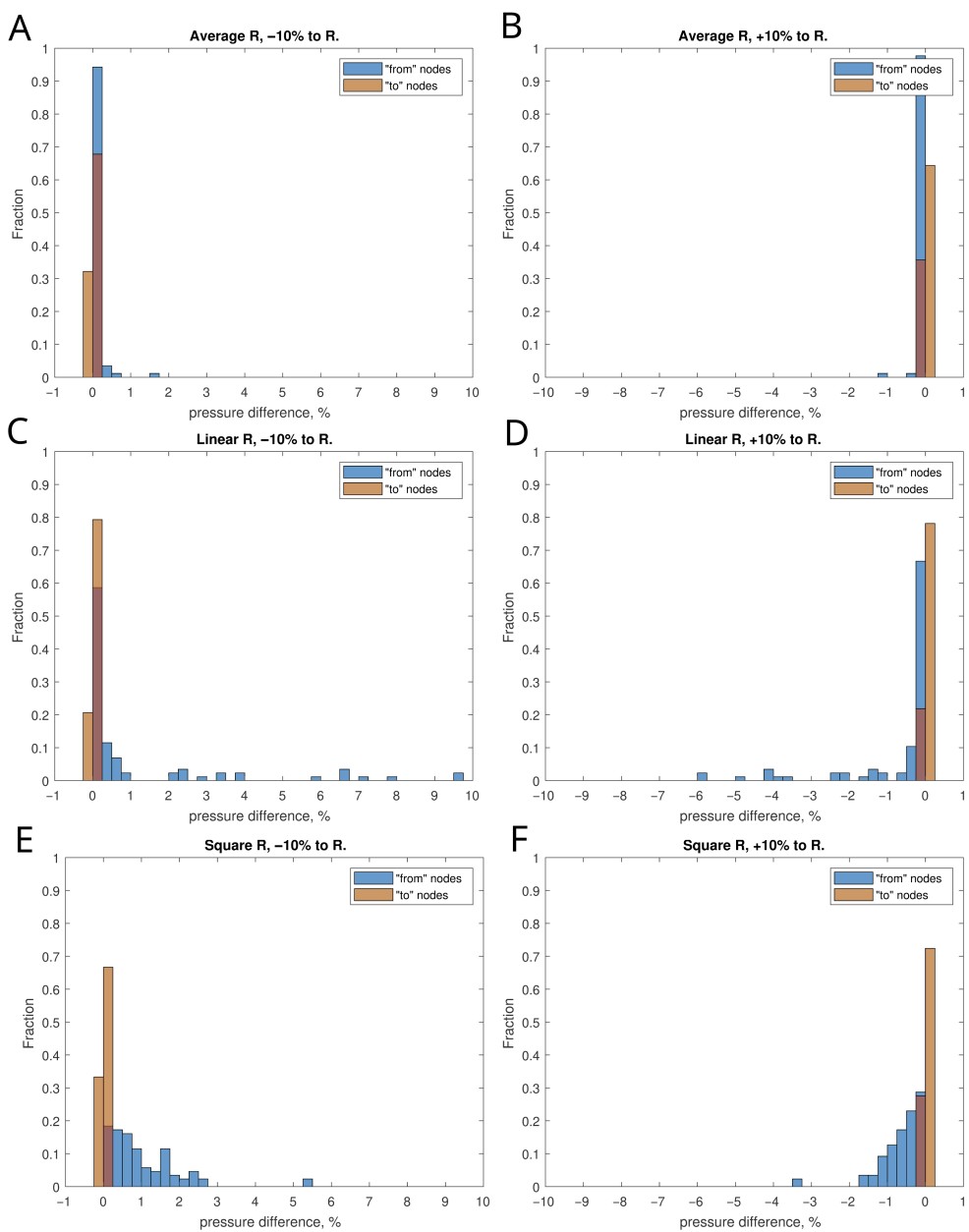

**Figure 5.** The distributions of the changes of pressures in the lymphatic vessels of the LS (vertices of the graph) after variation of the length diameters (±10%) for Scenario 1 (**A,B**), Scenario 2 (**C,D**), and Scenario 3 (**E,F**).

## 5. Topological Properties of the LS Graph

Following our previous study of the human LS [18], we characterized the topological properties of the murine LS graph model using some fundamental metrics. The quantified topological properties of the LS graph characterize the following structural and physiological features of the lymphatic network: the maximum number of lymphatics vessels entering and leaving a LN (maximum degree), the length of the shortest closed path (girth), the maximum distance between maximally separated nodes (diameter), the minimum distance between maximally separated nodes (radius), the typical distances between nodes (average path length), the characteristic of network regularity (energy and the spectral radius), characteristic of the network sparsity (edge density), the measure of how many nodes cluster together (clustering coefficient), the number of nodes critical for connectivity of the LS (number of separators), and the characteristics of lymph flow diversity (topological diversity of the vertices).

Let $G = (n, m)$ be the graph with $n$ nodes and $m$ edges, respectively. Consider the following characteristics:

- The number of input nodes $N_{inp}$, i.e., the number of nodes with degree 1 and out-degree 0;
- Maximum degree of graph $\Delta_G$, i.e., the maximum degree of its vertices;
- Girth of the graph $g$, which is the length of the shortest cycle in the simple graph;
- Diameter, i.e., the longest geodesic distance (in other terms, maximum eccentricity of any vertex)

$$D = \max_{v \in V} \epsilon(v) = \max_{v \in V} \max_{u \in V} d(u, v), \tag{6}$$

  where $d(u, v)$ is the geodesic distance (shortest oriented path connecting vertices $u$ and $v$), $\epsilon(v)$ is the eccentricity of vertex $v$;

- Radius of the graph (minimum eccentricity of any vertex),

$$r = \min_{v \in V} \epsilon(v) = \min_{v \in V} \max_{u \in V} d(u, v); \tag{7}$$

- Average path length (mean geodesic distance)

$$l_G = \frac{1}{n(n-1)} \sum_{u,v \in V,\, u \neq v} d(u, v); \tag{8}$$

- The energy and the spectral radius of the graph are defined as follows,

$$En(A) = \sum_{j=1}^{n} |\lambda_j|, \quad \rho(A) = \max\{|\lambda_j|\}, \tag{9}$$

  where $\lambda_j$ stand for the eigenvalues of the adjacency matrix $A$ of the graph;

- Edge density of the graph, i.e., the number of edges divided by the number of all possible edges,

$$\rho_d = \frac{m}{n(n-1)}; \tag{10}$$

- The clustering coefficient $C$ (transitivity) measures the probability that two neighbors of a vertex are connected. It can be computed as function of adjacency matrix $A$:

$$C(A) = \frac{\sum_{i=1,j=1,k=1}^{n,n,n} a_{ij} \cdot a_{jk} \cdot a_{ki}}{\sum_{i=1}^{n} \left( \left( \sum_{j=1}^{n} a_{ij} \right) \cdot \left( \left( \sum_{j=1}^{n} a_{ij} \right) - 1 \right) \right)}; \tag{11}$$

- Number of separators $n_{sep}$, i.e., the vertices removal of which disconnects the graph;

- Topological diversity of the vertices as a function of the Shannon entropy associated with flow rates through the incident edges,

$$D_{flow}(v_i) = \frac{H(v_i)}{\log(k)} = \frac{-\sum_{j=1}^{k} p_{ij} \log(p_{ij})}{\log(k)}, \quad p_{ij} = \frac{|Q_{ij}|}{\sum_{j=1}^{k} |Q_{ij}|}, \tag{12}$$

where $k$ is the number of $v_i$'s incident edges and $p_{ij}$ is the proportion of the flow between the adjacent $v_i$ and $v_j$ to the total flow through the edges involving $v_i$. The flow diversity is defined similar to the definition of network diversity in [33].

To analyze the robustness of the mouse LS to damage, we sequentially removed individual nodes of the graph and checked how many source vertices remained connected to the sink vertex into the circulatory system. The subgraphs of the LS and whole graph were analyzed. Accordingly, the robustness of the graph was estimated as the arithmetic mean of the ratio of the number of source vertices that retained the connection to the sink to their total number in the graph/subgraphs.

The summary of topological properties of the murine LS graph model are presented in Table 2. The characteristics of the whole LS graph and the two subgraphs representing the LS parts draining the draining the left ∪ low and right parts of the body are specified.

**Table 2.** Summary statistics for the anatomy-based graph of murine lymphatic system.

| Property | Whole Graph, $g$ | Left Subgraph, $g_l$ | Right Subgraph, $g_r$ |
|---|---|---|---|
| $G(n, m)$ | $g(88, 87)$ | $g_l(61, 60)$ | $g_r(27, 27)$ |
| Number of inlet vertices | 52 | 36 | 16 |
| Maximum degree | 5 | 5 | 5 |
| Girth | 3 | 0 | 3 |
| Diameter, oriented (simple) | 7 (11) | 7 (11) | 4 (7) |
| Radius | 4 | 6 | 4 |
| Average path length, dir. (undir.) | 2.5 (5.3) | 2.7 (5.5) | 1.9 (3.9) |
| Energy | 95.1 | 65.6 | 29.5 |
| Spectral radius | 2.93 | 2.9 | 2.93 |
| Edge density | 0.0114 | 0.0164 | 0.0385 |
| Clustering coefficient | 0.019 | 0 | 0.059 |
| Number of separators | 36 (in total) | 25 | 11 |
| Robustness | 0.917 | 0.863 | 0.825 |
| Average topological flow diversity, | | | |
| 　　　　- scenario 1: | 0.8252 | 0.8028 | 0.9039 |
| 　　　　- scenario 2: | 0.8242 | 0.8028 | 0.9015 |
| 　　　　- scenario 3: | 0.8242 | 0.8028 | 0.9023 |
| Number of LNs | 27 | 19 | 8 |

## 6. Conclusions

In this work, we have developed a graph model of the LS network in mice. To define directions of edges in the original anatomy-based simple graph, we considered the mass balance of global lymph flow in the LS. The local interstitial pressure was not considered for the final definition of the oriented graph. It is the first study in which a geometric model of the murine LS has been developed and characterized in terms of its structural organization and the lymph transfer function. The study complements our previous analysis of the human lymphatic system [18]. The developed graph model of the LS in mice provides a computational tool for studying the spatial aspects of the immune system functioning. It goes in line with recent advances in experimental techniques to characterize the whole-body dynamics of systemic infections in experimental mice [34].

The predicted orientation of the edges in the graph deserves further biological verification. To this end, various intravital imaging techniques could be used, e.g., near-infrared fluorescence imaging of lymphatic drainage patterns in mice [35], indocyanine green lymphangiography, or Doppler optical coherence tomography [4], to name just few of them [6].

We have previously implemented a similar approach to the analysis and modeling of the network structure of the human LS. The mouse and human LS differ fundamentally in terms of their cardinality, i.e., the number of elements comprising the system. The LS of mice consists of 28–36 LNs, whereas their number in the human LS ranges from about 500 to 1000. As a direct consequence, the graph model of the human LS is characterized by a much larger variability and more prominent randomness in its structure. The estimates of the probabilities of a new edge creation $P_e$ and the edge to connect nodes of different layers $P_o$ in random graph approximation of the LS quantify a larger uncertainty in the structure of the human LS graph compared to the mouse LS graph, i.e., 0.035 vs. 0.851 and 0.21 vs. 0.66, respectively, [18,36]. In addition, the diameter, radius, average path length, and energy features of the respective LS graphs differ substantially. However, the topological properties of the graph models, such as the girth, spectral radius, edge density, and clustering coefficients are close for both human and mouse LSs.

The graph scheme of the mouse LS formulated to study the search time of antigen presenting cells by T cells in the LS has been recently presented in [36]. However, the essential details of the LS, such as the adjacency matrix, the orientation of the edges, and the lymph flow estimates are not provided. In our study, we present a systematic analysis of the available anatomical and physiological data to develop the network model of the mouse LS, quantify the topological properties of the LS graph, and calculate flow through the network after making a series of assumptions about vessel diameters and terminal pressures.

The estimated parameters of the LS function in terms of the lymph flow rate and transfer time between various parts of the mouse body can be used in compartmental modeling for evaluation of the pharmacokinetic characteristics of drugs and adoptive cell therapies in advance of experiments. A remarkable example of the use of our recently developed graph model of human LS [18] is given in the study of how the topology of the lymphatic network affects the time required for an immune search through the lymphatic network to be completed [36].

To use the Hagen–Poiseuille equation for lymph flow, we assumed that the lymph is incompressible and Newtonian, the flow is laminar, and the vessels have constant circular cross-section with their length longer than the vessel diameter. However, the actual physiology of lymph flow through the LS is not considered, such as the lymphangion structure of the vessels, the pumping due to active contraction of the lymphatic muscle cells, adjacent tissue movement, and passive behavior properties of the vessels [5,14].

Further development of the presented graph model of the LS can be envisioned to proceed along three lines:

- Considering the biomechanics of lymphatic pumping through a chain of lymphangions and lymph nodes;
- Coupling the LS model with the cardiovascular system;
- Integration with multi-physics models of the immune system.

The explored scenarios of lymph vessel radii reflect three different modalities of LS network construction. The computational results predict how the structural parameters impact the functional properties of the LS, such as lymph flow velocity and the transfer time between the nodes. These contribute to a better understanding of the LS in health and disease [6]. Overall, our study meets the demand for quantitative rigorous approaches in the growing field of immunoengineering to utilize or exploit the lymphatic system for immunotherapy first in experimental animals and then to cure human immune-dependent diseases [4,37,38].

**Supplementary Materials:** The following supporting information can be downloaded at: https://www.mdpi.com/article/10.3390/a16030168/s1.

**Author Contributions:** Conceptualization, D.G., R.S., G.L. and G.B.; methodology, D.G., R.S. and G.B.; software, D.G., R.S. and E.Z.; validation, D.G., R.S., E.Z., G.L. and G.B.; formal analysis, D.G., R.S. and E.Z.; investigation, D.G., R.S., E.Z., G.L. and G.B.; data curation, G.L.; writing—original draft

preparation, D.G., R.S. and G.B.; writing—review and editing, D.G., G.L. and G.B.; visualization, D.G. and R.S.; supervision, G.B.; funding acquisition, G.B. All authors have read and agreed to the published version of the manuscript.

**Funding:** This research was funded by the Russian Science Foundation grant number 18-11-00171. R.S., D.G., and G.B. were partly supported by the Russian Foundation for Basic Research grant number 20-01-00352 and by Moscow Center for Fundamental and Applied Mathematics (agreement with the Ministry of Education and Science of the Russian Federation No. 075-15-2022-286).

**Data Availability Statement:** The developed directed graph model of the murine lymphatic system is contained within the Supplementary Material.

**Conflicts of Interest:** The authors declare no conflict of interest. The funders had no role in the design of the study; in the collection, analyses, or interpretation of data; in the writing of the manuscript, or in the decision to publish the results.

## Abbreviations

The following abbreviations are used in this manuscript:

LS    Lymphatic system
LN    Lymph node

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
