# Peer review of "Network Modeling of Murine Lymphatic System"

_algorithms, doi:10.3390/a16030168_

Round 1
Reviewer 1 Report
The manuscript "Network modelling of murine lymphatic system" is a valuable contribution to the field of immunology research, particularly in the context of animal models of disease. The authors present a novel approach to modeling the geometry and network structure of the murine lymphatic system (LS) using anatomical data and a mass balance approach to lymph dynamics based on the Hagen-Poiseuille equation.
The study is the first to develop a geometric model of the murine LS and to characterize it in terms of its structural organization and lymph transfer function. The results provide a computational tool for studying the spatial aspects of the immune system functioning in mice, which can be used to evaluate pharmacokinetic characteristics of drugs and adoptive cell therapies in advance of experiments.
The authors also discuss potential further development of the presented graph model of the LS, including considering the biomechanics of lymphatic pumping through a chain of lymphangions and lymph nodes, coupling the LS model with the cardiovascular system, and integration with multi-physics models of the immune system.
Overall, this study meets the demand for quantitative mechanistic approaches in immunoengineering to exploit the lymphatic system for immunotherapy in experimental animals and potentially in human immune-dependent diseases.
Author Response
Reviewer 1 stated that “The manuscript "Network modelling of murine lymphatic system" is a valuable contribution to the field of immunology research, particularly in the context of animal models of disease. The authors present a novel approach to modeling the geometry and network structure of the murine lymphatic system (LS) using anatomical data and a mass balance approach to lymph dynamics based on the Hagen-Poiseuille equation…”
We thank the Reviewer for insightful comments and the thorough work on our manuscript.

Reviewer 2 Report
Flow through the lymphatic system plays a critical role in immune responses and tissue fluid homeostasis, but it has received little attention, compared with the cardiovascular system. Here, the authors construct a graph of the lymphatic system of mice and calculate flow through the network after making a series of assumptions about vessel diameters and terminal pressures. The study is simple, but novel –– and potentially useful for physiologists studying the systemic transport of lymph in mice. It also is generalizable to other network analyses. There are a few issues that should be addressed to improve the presentation:
1) The authors should explain the difference between "oriented" and "directed"; if they are the same, then use consistent wording.
2) The authors should discuss any similarities or differences between the mouse and human systems analyzed with this approach.
3) In Figure 3, why was scenario 1 (constant vessel diameters) chosen? Do the authors believe that scenario 1 was the most likely (or the most physiological)?
4) In section 5, the authors measure a number of topological properties of the LS graph, but with little explanation. Some rationale should be included as to why these are important, and how they might be physiologically -relevant.
5) The authors applied a digraph for the lymphatic system. They need to explain how they define directions of edges, is this just based on mass balance or local interstitial pressure is also considered for the direct definition of the digraph.
6) Two outlet vertices (red circles 28 and 29) and multiple inlet vertices (yellow circles) have been defined as sinks and sources of the lymphatic system. They set equal pressure at the two sink nodes and equal velocity at the inlet vortices. The authors need to explain why they set all the inflows the same, or mention this simplification as a limitation for their computational model. An alternative approach would be to define proportional inflows based on the local impedance of the related edges.
7) The authors have not elaborated the major aspects of their computational model including numerical algorithm and assumptions/limitations. It would be useful to explain how this digraph can be biologically verified.
8) the introduction is very short and should be expanded to include a discussion of previous modeling work in the context of lymphatic networks.
Minor:
9) There is a missing reference in line 174: "The flow diversity is defined similar to the definition of network diversity in [? ] "
10) The authors should explain the unique fluid dynamics relevant in the lymphatic system and also mention the assumptions that allow you to use Hagen-Poiseuille equation for lymphatic flow as the limitation/simplifications of the model.
11) In figure 1: forelimb and hindlimb do not exist in the definition table.
12) In figure 2: All the axes should be declared in the caption; what are vi, vj, …?
13) In table 1, the values of some parameters are presented with ranges. It would be useful to add another column with the exact value used for the computational results.
Author Response
We thank the Reviewer for insightful comments and the thorough work on our manuscript.
Reviewer 2 stated that “The study is simple, but novel –– and potentially useful for physiologists studying the systemic transport of lymph in mice. It also is generalizable to other network analyses. There are a few issues that should be addressed to improve the presentation.”
Our reply:
All the issues have been addressed in the revised manuscript as described below. The corrections/extensions are marked in red.
(1) The authors should explain the difference between "oriented" and "directed"; if they are the same, then use consistent wording.
Our reply
In the present study, the use of the notion of an oriented graph is appropriate. We have corrected the wording across the whole manuscript and added the following explanation to the text of Section 2:
“A simple graph G=(V,E) is a pair of sets V and E, with elements of V being vertices or nodes and E being edges (Diestel, 2017). The simple graph with edges oriented in only one direction is called an oriented graph.”
Diestel Reinhard (2017). Graph Theory (5th ed.), Springer Berlin, Heidelberg. ISBN. 978-3-662-53621-6, Number of Pages XVIII, 428. doi https://doi.org/10.1007/978-3-662-53622-3
(2) The authors should discuss any similarities or differences between the mouse and human systems analyzed with this approach.
Our reply:
In response to the suggestion we have added the following discussion to Conclusions section.
“We have previously implemented a similar approach to analysis and modelling of the network structure of human LS. The mouse and human LS differ fundamentally in terms of their cardinality, i.e. the number of elements comprising the system. The LS of mice consists of 28-36 LN, whereas their number in human LS ranges from about 500 to 1000. As a direct consequence, the graph model of the human LS is characterized by a much larger variability and more prominent randomness in its structure. The estimates of the probabilities of a new edge creation Pe and the edge to connect nodes of different layers Po in random graph approximation of the LS quantify a larger uncertainty in the structure of the human LS graph compared to the mouse LS graph, i.e. 0.035 vs 0.851 and 0.21 vs 0.66, respectively (Savinkov et al., 2020; Fredous et al., 2022). In addition, the diameter-, radius-, average path length- and energy features of the respective LS graphs differ substantially. However, the topological properties of the graph models, such as the girth, spectral radius, edge density and clustering coefficients are close for human and mouse LS.”
(3) In Figure 3, why was scenario 1 (constant vessel diameters) chosen? Do the authors believe that scenario 1 was the most likely (or the most physiological)?
Our reply:
In response to the comment we have now displayed the results of Scenario 2 as more physiological and added the following information:
“For human LS it is known that largest lymphatic vessels have a diameter about 2 mm and the diameter reduces to approximately 10-60 μm for initial lymphatic capillaries (Margaris, & Black, 2012).”
and modified the justification to read
“Due to the lack of detailed anatomical and physiological data on the diameters of all lymphatic vessels in mice, we explored three complementary assumptions on the distribution of the radii of the edges of the lymphatic graph,…”
(4) In section 5, the authors measure a number of topological properties of the LS graph, but with little explanation. Some rationale should be included as to why these are important, and how they might be physiologically -relevant.
Our reply:
In response to the above request, we have added the following explanation of their meaning and relevance.
“The quantified topological properties of the LS graph characterize the following structural and physiological features of the lymphatic network: the maximum number of lymphatics vessels entering and leaving a LN (maximum degree), the length of the shortest closed path (girth), the maximum distance between maximally separated nodes (diameter), the minimum distance between maximally separated nodes (radius), the typical distances between nodes (average path length), the characteristic of network regularity (energy and the spectral radius), characteristic of the network sparsity (edge density), the measure of how many nodes cluster together (clustering coefficient), the number of nodes critical for connectivity of the LS (number of separators), and the characteristic of lymph flow diversity (topological diversity of the vertices).”
(5) The authors applied a digraph for the lymphatic system. They need to explain how they define directions of edges, is this just based on mass balance or local interstitial pressure is also considered for the direct definition of the digraph.
Our reply:
The directions of edges are defined via computation of lymph flows in the LS with Hagen-Poiseuille equation. This is explained in details in subsection 3.1. The local interstitial pressure is not considered as there is no empirical data for that now. We added a corresponding sentence in Conclusion section:
“To define directions of edges in the original anatomy-based simple graph, we considered the mass balance of global lymph flow in the LS. The local interstitial pressure was not considered for the final definition of the oriented graph.”
(6) Two outlet vertices (red circles 28 and 29) and multiple inlet vertices (yellow circles) have been defined as sinks and sources of the lymphatic system. They set equal pressure at the two sink nodes and equal velocity at the inlet vortices. The authors need to explain why they set all the inflows the same, or mention this simplification as a limitation for their computational model. An alternative approach would be to define proportional inflows based on the local impedance of the related edges.
Our reply:
In response to the comment, we have added the following clarification at the end of Section 2:
“The initial collecting lymphatics are likely to differ in the inflows as they absorb lymph from interstitial space characterised by volume and pressure varying across the body. However, the respective anatomical and physiological data to quantify the local impedance of the related edges are largely missing. Hence, we used a simplifying assumption that the flow velocity at all inlet vortices are the same and the pressure at the two sink nodes are equal.”
(7) The authors have not elaborated the major aspects of their computational model including numerical algorithm and assumptions/limitations. It would be useful to explain how this digraph can be biologically verified.
Our reply:
We have added the following details of the computational model at the end of subsection 3.1:
“The graph was constructed and visualized using the R package igraph. The algorithm for computing the flows described in Section 3.1 was implemented in R using the ginv() function for pseudo-inverse calculation from the MASS package. In addition, we have verified the computation of the flows in Julia using the singular value decomposition to obtain the pseudo-inverse.”
In response to the second comment, we have added to Conclusions section that
“The predicted orientation of the edges in the graph deserves further biological verification. To this end, various of intravital imaging techniques could be used, e.g., near-infrared fluorescence imaging of lymphatic drainage patterns in mice (Agollah et al 2014), indocyanine green lymphangiography, Doppler optical coherence tomography (O’Melia et al, 2019) to name just few of them (Singhal et al., 2023).”
Agollah GD, Wu G, Sevick-Muraca EM, Kwon S. In vivo lymphatic imaging of a human inflammatory breast cancer model. J Cancer. 2014 Oct 23;5(9):774-83. doi: 10.7150/jca.9835.
O'Melia MJ, Lund AW, Thomas SN. The Biophysics of Lymphatic Transport: Engineering Tools and Immunological Consequences. iScience. 2019 Dec 20;22:28-43. doi: 10.1016/j.isci.2019.11.005.
Singhal D, Börner K, Chaikof EL, Detmar M, Hollmén M, Iliff JJ, Itkin M, Makinen T, Oliver G, Padera TP, Quardokus EM, Radtke AJ, Suami H, Weber GM, Rovira II, Muratoglu SC and Galis ZS (2023), Mapping the lymphatic system across body scales and expertise domains: A report from the 2021 National Heart, Lung, and Blood Institute workshop at the Boston Lymphatic Symposium. Front. Physiol. 14:1099403. doi: 10.3389/fphys.2023.1099403
(8) the introduction is very short and should be expanded to include a discussion of previous modeling work in the context of lymphatic networks.
Our reply:
We have added the following paragraph summarizing the state of the art in modelling and analysis of lymphatics networks to Introduction section.
“The primary function of the lymphatic system is the maintenance of the interstitial fluid homeostasis (Margaris and Black, 2012). The structure and topology of LS network is heterogeneous and remains to be systematically explored (Moore and Bertram, 2018). The existing mathematical models of lymphatic system refer to specific parts of it, such as lymphatic capillary network (Roose and Swartz, 2012), collecting lymphatics (Bertram et al., 2018), lymphangions (Morris et al. 2021), or branching networks of lymphatic vessels (Jamalian et al., 2016). The computational models of the whole lymphatic system network are still rare (Reddy et al., 1977; Farooqi and Mohler, 1989; Mozokhina and Mukhin, 2018; Savinkov et al, 2020). Comprehensive reviews of existing approaches to modelling the lymphatic system structure and function can be found in (Margaris and Black, 2012; Mozokhina and Savinkov, 2020). One of the major bottleneck in developing the computational models of the lymphatic system is due to the lack of comprehensive anatomical and physiological data (Margaris and Black, 2012). Latest research activity has clearly stated that “Thus, gross lymphatic anatomy has not been updated for more than a century.our knowledge of macro-lymphatic anatomy remains rudimentary” (Singhal et al., 2023). The existing gaps (Hsu et al. 2016), require further systematic research (Hur et al. 2023) including mathematical modelling, the later serves to integrate available knowledge and to identify critical issues amenable to further biological testing.”
We have also added the following commentary at the end of the manuscript.
“The graph scheme of the mouse LS formulated to study the search time of antigen presenting cells by T cells in LS has been recently presented in (Ferdous et al. 2022). However, the essential details of the LS such the adjacency matrix, the orientation of the edges and the lymph flow estimates are not provided. In our study, we present a systematic analysis of the available anatomical and physiological data to develop the network model of the mouse LS, quantify the topological properties of the LS graph and calculate flow through the network after making a series of assumptions about vessel diameters and terminal pressures.”
Minor:
(9) There is a missing reference in line 174: "The flow diversity is defined similar to the definition of network diversity in [? ] "
Our reply:
The reference has been added.
(10) The authors should explain the unique fluid dynamics relevant in the lymphatic system and also mention the assumptions that allow you to use Hagen-Poiseuille equation for lymphatic flow as the limitation/simplifications of the model.
Our reply:
We have added the following comment to the Conclusion section:
“To use the Hagen-Poiseuille equation for lymph flow, we assumed that the lymph is incompressible and Newtonian, the flow is laminar and the vessels have constant circular cross-section with their length is longer than the vessel diameter. However, the actual physiology of lymph flow through the LS is not considered, such as the lymphangion structure of the vessels, the pumping due to active contraction of the lymphatic muscle cells, adjacent tissue movement and passive behavior properties of the vessels (Jamalian et al. 2016; Moore et al. 2018).”
(11) In figure 1: forelimb and hindlimb do not exist in the definition table.
Our reply:
We have included forelimb and hindlimb into the definition table.
(12) In figure 2: All the axes should be declared in the caption; what are vi, vj, …?
Our reply:
We have expanded the caption of Figure 2 with the following sentence:
"The i-th and j-th vertices and edges are denoted as v_i, v_j and e_i, e_j, respectively."
13) In table 1, the values of some parameters are presented with ranges. It would be useful to add another column with the exact value used for the computational results.
Our reply:
The values presented in Table refer characterize the anatomical and physiological properties of the lymphatic system coming from various sources. The set of parameters is specified in paragraph at the end of Section 2 as follows:
“The following values of model parameters were used in computations of lymph flows
- number of inlet vertices n_in = 52,
- vessel radii range r_ij = 40 – 300 μm
- vessel length l_ij = 7 - 60 mm
- pressure at the sink nodes p_out = 725 Pa,
- lymph viscosity μ = 1.81 mPa s,
- lymph inflow q_in = 0.4 mL/day.”

Reviewer 3 Report
I read the article with great interest.
I congratulate the authors on choosing the topic. Modeling biological processes is always a challenge that requires knowledge and the ability to use it to create a model.
In my opinion, the authors handled it very well. Their article is clear and well-written.
Comments:
Reference [20] is incorrectly given - it contains deficiencies; only the authors are listed, without the article's title and the journal's name. This should be supplemented.
On page 10, instead of references, there is [?]
I think it would be helpful to have a diagram in a biological context that would show people who are not familiar with the lymphatic system what the role of this system is in the organism. If not a diagram, then at least a short description in the Introduction.
I also have a question about scenarios. Does anything follow from them in a biological context?
Can any conclusions be drawn from them?
Author Response
We thank the Reviewer for insightful comments and the thorough work on our manuscript.
Reviewer 3 stated that “Modeling biological processes is always a challenge that requires knowledge and the ability to use it to create a model. In my opinion, the authors handled it very well.”
All the concerns have been addressed in the revised manuscript as described below.
Comments:
(1) Reference [20] is incorrectly given - it contains deficiencies; only the authors are listed, without the article's title and the journal's name. This should be supplemented.
Our reply:
It has been corrected (now appearing as [38]).
(2) On page 10, instead of references, there is [?]
Our reply:
It has been corrected (now appearing as [32]).
(3) I think it would be helpful to have a diagram in a biological context that would show people who are not familiar with the lymphatic system what the role of this system is in the organism. If not a diagram, then at least a short description in the Introduction.
Our reply:
In response to the above suggestion, we have added a short paragpga to the introduction:
“The lymphatic system is a body-wide network of lymphatic vessels and lymphoid organs with two major functions: (1) the fluid transport from tissues to the blood system to maintain the fluid homeostasis and (2) trafficking of antigens and immune cells to lymph nodes where the immune responses take place (O’Melia et al., 2019; Moore et al., 2018). Lymphoid organs include the large number of lymph nodes as well as spleen, thymus, tonsils and bone marrow (Moore et al., 2018). The lymphatic vessels are the conduits that facilitate the directional lymph transport from peripheral tissues to secondary lymph nodes (O’Melia et al., 2019). Understanding of the lymphatic structural and functional organization is essential to discern how LS interacts with different tissues and organs within the body (Singhal et al., 2023).”
O'Melia MJ, Lund AW, Thomas SN. The Biophysics of Lymphatic Transport: Engineering Tools and Immunological Consequences. iScience. 2019 Dec 20;22:28-43. doi: 10.1016/j.isci.2019.11.005.
Moore JE Jr, Bertram CD. Lymphatic System Flows. Annu Rev Fluid Mech. 2018 Jan;50:459-482. doi: 10.1146/annurev-fluid-122316-045259.
Singhal D, Börner K, Chaikof EL, Detmar M, Hollmén M, Iliff JJ, Itkin M, Makinen T, Oliver G, Padera TP, Quardokus EM, Radtke AJ, Suami H, Weber GM, Rovira II, Muratoglu SC and Galis ZS (2023), Mapping the lymphatic system across body scales and expertise domains: A report from the 2021 National Heart, Lung, and Blood Institute workshop at the Boston Lymphatic Symposium. Front. Physiol. 14:1099403. doi: 10.3389/fphys.2023.1099403
(4) I also have a question about scenarios. Does anything follow from them in a biological context?
Can any conclusions be drawn from them?
Our reply:
In response to above request, we have added the following comment to Conclusions section
“The explored scenarios of lymph vessel radii reflect three different modalities of LS network construction. The computational results predict how the structural parameters impact the functional properties of LS, such as lymph flow velocity and the transfer time between the nodes. These contribute to a better understanding of the LS in health and disease (Singhal et al. 2023).”
